# Pediatric High Grade Glioma Classification Criteria and Molecular Features of a Case Series

**DOI:** 10.3390/genes13040624

**Published:** 2022-03-31

**Authors:** Anna Maria Buccoliero, Laura Giunti, Selene Moscardi, Francesca Castiglione, Aldesia Provenzano, Iacopo Sardi, Mirko Scagnet, Lorenzo Genitori, Chiara Caporalini

**Affiliations:** 1Pathology Unit, Meyer Children’s Hospital, 50139 Florence, Italy; selene.moscardi@meyer.it (S.M.); chiara.caporalini@meyer.it (C.C.); 2Neuro-Oncology Unit, Department of Pediatric Oncology, Meyer Children’s Hospital, 50139 Florence, Italy; laura.giunti@meyer.it (L.G.); i.sardi@meyer.it (I.S.); 3Pathological Anatomy, Careggi Hospital, 50139 Florence, Italy; francesca.castiglione@gmail.com; 4Medical Genetics, Department of Experimental and Clinical Biomedical Sciences Mario Serio, University of Florence, 50139 Florence, Italy; aldesia.provenzano@unifi.it; 5Neurosurgery Unit, Meyer Children’s Hospital, 50139 Florence, Italy; mirko.scagnet@meyer.it (M.S.); lorenzo.genitori@meyer.it (L.G.)

**Keywords:** pediatric, high-grade glioma, astrocytoma, thalamic glioma, gene panel, *IDH2* mutation, *EZHIP*, *TP53*, *H3F3A*, case report

## Abstract

Pediatric high-grade gliomas (pHGGs) encompass a heterogeneous group of tumors. Three main molecular types (*H3.3* mutant, *IDH* mutant, and *H3.3/IDH* wild-type) and a number of subtypes have been identified. We provide an overview of pHGGs and present a mono-institutional series. We studied eleven non-related pHGG samples through a combined approach of routine diagnostic tools and a gene panel. *TP53* and *H3F3A* were the most mutated genes (six patients each, 54%). The third most mutated gene was *EGFR* (three patients, 27%), followed by *PDGFRA* and *PTEN* (two patients each, 18%). Variants in the *EZHIP*, *MSH2*, *IDH1*, *IDH2*, *TERT*, *HRAS*, *NF1*, *BRAF*, *ATRX*, and *PIK3CA* genes were relatively infrequent (one patient each, 9%). In one case, gene panel analysis documented the presence of a pathogenic *IDH2* variant (c.419G>A, p.Arg140Gln) never described in gliomas. More than one-third of patients carry a variant in a gene associated with tumor-predisposing syndromes. The absence of constitutional DNA did not allow us to identify their constitutional origin.

## 1. Introduction

Central nervous system (CNS) tumors are the most common solid neoplasms in childhood [1]. Among these, approximately 50% are gliomas [1]. Unlike in adults, low-grade gliomas (LGGs) predominate in children, while high-grade gliomas (HGGs) are less frequent [2,3]. The incidence of pediatric HGGs (pHGGs) has been calculated as 1.1–1.78 per 100,000 children [4]. Despite their low incidence, pHGGs are responsible for over 40% of all childhood brain tumor death, and overall, they are the more common cause of tumor-related death at this age [1,3]. Although the morphological features of pHGGs are comparable to those of HGGs in adulthood, the genetic–molecular characteristics differ so much that some of the novel therapeutic strategies derived from research on adult gliomas have yielded unsatisfactory results [5]. In addition, while adult HGGs are more often restricted to the cerebral hemispheres, those affecting children can occur throughout the central nervous system, with about half of cases occurring in midline locations [4]. Important molecular differences also exist within pHGGs. In this regard, on the basis of the genomic, epigenomic and transcriptomic profile, three main molecular types and a series of molecular subtypes—each with clinical peculiarities (i.e., age, tumor location, prognosis, and potential targetable therapy)—have been identified in the current scientific publications [4,5,6]. The three main molecular types of pHGGs are the histone H3 mutant, the isocitrate dehydrogenase gene *(IDH)* mutant, and the *H3/IDH* wild-type [4,5,6]. Additionally, there are the infant-type hemispheric gliomas and diffuse-midline glioma epidermal growth factor receptor (*EGFR*) mutant [7,8]. The molecular stratification of pHGGs into types and subtypes is essential due to the availability of promising new targeted therapies.

### 1.1. Histone H3 Mutant pHGGs

About 40% of pHGGs fall into the subgroup in which the genes encoding for histone H3 are mutated [5]. H3 is one of the major families of histones. Histones play an essential role in the condensation of DNA by contributing to the formation of nucleosomes. They may undergo post-translational modifications (i.e., methylation, acetylation and phosphorylation) that modify their interaction with DNA and alter a number of biological processes such as gene expression, DNA repair, mitosis and meiosis [9].

The majority of H3 mutant pHGGs harbor variants at position 27 consisting of a substitution of lysine with methionine (Lys27Met) in the H3.3 (*H3F3A*) gene (about three-fourths) or in H3.1 (*HIST1H3B/C*) gene (about one-fourth). The consequence of these variants is the biochemical inhibition of Polycomb Repressor Complex 2 (PRC2) (a protein exhibiting histone methyltransferase activity) through the sequestration of its catalytic subunit Enhancer of Zeste Homolog 2 (EZH2). The result is a global loss of trimethylation of lysine 27 (H3.3K27me3) on all H3 molecules both mutated and wild type [10]. The inhibition of the methylation pathways promotes tumorigenesis through several modalities, particularly by altering of the dynamics of chromatin. H3 mutant tumors typically arise in the midline (pons, midbrain, thalamus, spinal cord) and usually affect school-aged children. H3.3 mutant HGGs have a very dismal prognosis despite the fact that the histological features may sometimes be consistent with LGGs. H3.1 mutant HGGs seem to respond slightly better to radiotherapy with a consequent less aggressive course [11]. It is debated whether the dismal prognosis of Lys27Met mutated gliomas is related to the midline location—which prevents surgical resection in most cases—or it is an independent prognostic factor. Indeed, a number of low-grade and non-midline tumors carry this variant without influencing prognosis [12,13].

*H3F3A* may also be mutated at position Gly34 due to the substitution of glycine with arginine or valine (H3.3G34R/V). Gly34Arg/Val results in a total reduction of H3K36me2 and H3K36me3 levels [9,14]. Moreover, Gly34Val affects the regulation of gene expression and results in up-regulation of *MYCN* [15]. An additional molecular feature of H3G34 mutant gliomas is the high frequency of O6-methylguanine DNA methyltransferase gene (*MGMT*) promoter methylation. Tumor-harboring Gly34Arg/Val variants are typically hemispheric and, when compared with H3.3K27M mutated gliomas, affect older age group (teenagers and young adults); they are associated with a slightly prolonged survival, possibly related to *MGMT* methylation, resulting in enhanced responsiveness to temozolomide [1,16,17].

### 1.2. IDH Mutant pHGGs

*IDH* genes, which encode for IDH enzymes, are frequently mutated in LGGs and secondary glioblastomas. IDH proteins are involved in the Krebs cycle, catalyzing the oxidative decarboxylation of isocitrate to α-ketoglutarate (α-KG). Variants in *IDH* inhibit the physiologic activity of the IDH proteins in converting isocitrate to α-KG, instead determining its conversion to 2-hydrosxyglutarate (2-HG) [18]. All variants result in an approximately 100-fold increase of 2-HG which is an oncometabolite that contributes to tumor development. Moreover, the IDH mutated proteins upregulate vascular endothelial growth factor (VEGF) and result in high levels of hypoxia-inducible factor-1α (HIF-1α), which can both ultimately contribute to glioma genesis [18]. Among the *IDH* family genes, *IDH1* and *IDH2* are the most frequently mutated in gliomas. Point variants of *IDH1* are by far the most frequent. *IDH1* and *IDH2* variants are heterozygous, of somatic origin, and mutually exclusive (cases of concurrent *IDH1* and *IDH2* variants are exceedingly rare) [19]. *IDH1* variants are strongly associated with astrocytomas *TP53* and alpha thalassemia/mental retardation syndrome X-linked (*ATRX*) mutated, while *IDH2* variants predominantly occur in oligodendrogliomas ATRX wild type and 1p-19q codeleted. Approximately 90% of *IDH1* variants are located in exon 4 at codon 132 where, in the majority of the cases, a CGT–CAT transition changes a single amino acid from arginine to histidine (Arg132His). In the few remaining cases, different variants have been described but are usually restricted to codon 132, which is the isocitrate-binding pocket of *IDH1*. *IDH2* variants are exclusively detected in arginine at position 172, which is the analogous site to arginine 132 in *IDH1*. Both low- and high-grade gliomas may harbor *IDH* variants. However, when cyclin-dependent kinase inhibitor 2A/B *(CDKN2A/B*) homozygous deletion co-occurs, the highest grade (grade 4) is assigned [6]. *IDH* variants are typically observed in adulthood, whereas they are rare at pediatric age [19]. Contrary to adult cases, *IDH*-mutated pHGGs do not show evidence of lower-grade precursor lesions. *IDH*-mutant pHGGs typically affect cerebral hemispheres and, similarly to what it observed in adults, have more favorable behavior and are likely to be associated with *MGMT* methylation, making temozolomide a potentially effective treatment option. Many pediatric *IDH*-mutated gliomas do not harbor the Arg132His with high incidence of rare variants such as Arg132Gly, Arg132Ser and Arg132Cys [2,20,21].

### 1.3. H3/IDH Wild Type pHGGS

The third molecular type of pHGG corresponds to *H3*/*IDH* wild-type lesions. This type is highly heterogeneous, comprising a number of largely hemispheric tumors affecting infants/newborns, children and adolescents, and displaying a variety of genomic and epigenetic features as well as different clinical behaviors. In particular, in this third molecular type fall pleomorphic-xantoastrocytoma (PXA)-like pHGGs, v-myc myelocytomatosis viral-related oncogene (*NMYC*)*,* platelet-derived growth factor receptor A (*PDGFRA*), or *EGFR* amplified gliomas and hypermutant pHGGs [5,22].

### 1.4. PXA-like pHGGs

PXA-like pHGGs are hemispheric lesions molecularly characterized by the v-raf murine sarcoma viral oncogene homolog B1 (*BRAF*) Val600Glu variant, usually along with *CDKN2A/B* gene deletion [6]. The *BRAF* gene is a proto-oncogene coding a protein also called BRAF. BRAF is a member of the rapidly accelerated fibrosarcoma (RAF) kinase family, which transduces signals downstream of the rat sarcoma viral oncogene homolog (RAS) via the mitogen-activated protein kinase (MAPK) pathway, playing a role in cell growth [23]. Many variants in the *BRAF* gene have been identified in cancer. These variants lead to extracellular signal-regulated kinase (ERK) activation. The val600Glu variant, consisting of the substitution of valine to glutamic acid at position 600, is the commonest. The *CDKN2A/B* tumor suppressor genes encode for p16 and p15 proteins. p16 and p15 inhibit cyclin-dependent kinase proteins, thus activating the retinoblastoma protein family with a consequent block of the transition from the G0-phase to the S-phase of the cell cycle. In CNS, both high-grade and low-grade tumors may carry Val600Glu and *CDKN2A/B* deletion. PXAs and gangliogliomas have the highest incidence of Val600Glu. PXA-like pHGGs have a better prognosis, even though they exhibit a high rate of recurrence [5,23,24].

### 1.5. pHGGs Amplified in NMYC, PDGFRA or EGFR

pHGGs amplified in *NMYC, PDGFRA* and *EGFR* may arise in children of any age and represent distinct molecular sub-types associated with significantly different behavior: *MYCN*-amplified gliomas show the poorest prognosis; *PDGFRA*-amplified gliomas, designated as RTK1 pHGGs, show a better prognosis; and the *EGFR*-amplified gliomas, designated RTK2 pHGGs, show an intermediate prognosis between the previous two [22].

*MYCN* is a member of the *MYC* family of oncogenes. It is highly expressed in neural tissue during embryogenesis. It encodes a basic helix-loop-helix–leucine zipper (bHLH-LZ) protein called N-Myc or MYCN. This protein plays a role in the regulation of gene transcription. The amplification of the *MYCN* gene is associated with a variety of tumors, most notably neuroblastoma, in which the level of amplification is related to a dismal prognosis. A small number of high-grade brain tumors, quantified in 8–9% of cases, may disclose *MYCN* amplification [22,25,26].

*PDGFRA* and *EGFR* encode for high-affinity cell surface receptors belonging to the receptor tyrosine kinase (RTKs) family. They are involved in important physiologic cellular processes. RTKs are generally activated by the interaction of receptor-specific RTK ligands. Alterations in genes encoding RTKs lead to interference in a series of signaling pathways involved in determining oncogenesis and neoplastic progression. *PDGFRA* is essential for glial development and is involved in gliomagenesis through different mechanisms such as *PDGFRA* amplification or activating variants [3,27]. Alterations in PDGF-driven signaling are prevalent in the majority of pediatric tumors, while EGF-driven signaling is predominant in adults [28,29].

### 1.6. Hyper Mutant pHGGs

Hyper-mutant pHGGs affect children of any age, are more often hemispheric, and are associated with a poor prognosis [5]. These tumors are characterized by microsatellite instability (MSI). Microsatellites are short, repeated segments of DNA (mononucleotide, dinucleotide, or higher-order nucleotide repeats). MSI mostly depends on the presence of inactivating variants in the DNA mismatch repair (MMR) genes, mainly mutL homologue 1 (*MLH1*), mutS homologue 2 (*MSH2*), mutS homologue 6 (*MSH6*), and postmeiotic segregation increased 2 (*PMS2*). The resulting accumulation of variants can lead to oncogenesis and neoplastic progression. MSI has been described in HGGs with a frequency ranging from a percentage close to 0% to more than 40%, probably as consequence of the different sensitivities of the method used to ascertain the MSI status and of the age of the patients studied [30]. In fact, MSI-related gliomas are more frequent in children than in adults. Variants in MMR genes may be sporadic or inherited as a monoallelic germline variant (Lynch syndrome) or biallelic variant (constitutional mismatch repair-deficiency, CMMRD) [31]. Moreover, MMR-acquired variants or overexpression correlate with temozolomide (TMZ) resistance in glioblastomas [32,33,34].

### 1.7. Infant-Type Hemispheric Gliomas

Infant-type hemispheric gliomas are rare pathological entities, clinically distinct from gliomas, that arise in older children. In fact, they are associated with a better outcome regardless of the histological grade (these tumors may encompass a wide range of histological features from low-grade to frankly high-grade morphology), even with incomplete surgical resection and the age-related impossibility of radiotherapy. The most common molecular alterations observed in HGGs occurring in infants are fusion events, particularly those involving neurotropic receptor tyrosine Kinase (*NTRK*)*1/2/3*. *NTRK1/2/3* encodes for tropomyosin receptor kinases (Trk) A, B and C, respectively [7]. Trk proteins belong to a family of growth factor receptors. They regulate synaptic strength and plasticity in the mammalian nervous system. The fusion genes resulting from the juxtaposition of the C-terminal kinase domain of *NTRK1/2/3*, with the N-terminal sequences of different genes, lead to the transcription of chimeric Trk proteins with oncogenic potential. *AGBL carboxypeptidase 4 (AGBL4)**4:NTRK2*, *tropomyosin 3*
*(TPM3):NTRK1*, and *ETS variant transcription factor 6 (ETV6):NTRK3* fusions are the more frequent fusions observed in infantile HGGs. Other fusion genes in infants involve anaplastic lymphoma kinase (*ALK)*, c-ros oncogene 1 (*ROS),* and mesenchymal-epithelial transition (*MET); ALK* is more often involved in LGGs, and *ROS* and *MET* in HGGs. The availability of anti-Trk drugs represents an interesting therapeutic opportunity for these patients [7].

### 1.8. Diffuse Midline Gliomas EGFR Mutant

Thalamic gliomas—mainly bilateral at the onset, showing *EGFR* variants, *EZHIP* (ex *CXorf67*) overexpression, and loss of H3.3K27me3 in the absence of *H3F3A* variants—have recently been described. They are rare and typically affect young children. Due to the impossibility of an effective surgical treatment, the prognosis is invariably poor irrespective of histological grade. Recent studies have documented that in the diffuse midline glioma *EGFR* mutant, PRC2 inhibition and the consequent H3.3K27me3 loss is related to an epigenetic aberrant overexpression of the EZH2 inhibitory protein EZHIP, instead of to *H3F3A* variants. The oncoprotein EZHIP and H3.3K27M are competitive inhibitors of PRC2 [35,36,37]. Considering that H3.3K27me3 loss is the main molecular feature of both the H3.3K27M mutant and diffuse midline *EGFR* mutant gliomas, it has been proposed that these tumors be grouped under the denomination of H3.3K27-altered pHGGs [36,38].

### 1.9. Aim

We present a mono-institutional series of pHGGs operated at the Surgical Unit of the Meyer Children’s Hospital of Florence. Our aim was to evaluate the presence of a number of further molecular alterations in addition to those evaluated for strictly diagnostic purposes.

## 2. Patients and Methods

### 2.1. Patients

Eleven non-related consecutive pHGGs for which fresh material was available for molecular study were used in the study. Six (55%) patients were female, and five (45%) were male. The mean age at diagnosis was 7 years (range 2 months–18 years). Five lesions were hemispheric (45%), two were thalamic (18%) and the remaining four tumors, respectively, arose from the midbrain, hypothalamus, cerebellum and spinal cord (9% for each one) (Table 1).

### 2.2. Methods

The original slides were re-evaluated according to the latest guidelines of the World Health Organization (WHO) Classification of CNS tumors [38]. Moreover, in order to molecularly characterize the tumors, a combined approach of routine diagnostic tools (immunohistochemistry: formalin-fixed and paraffin-embedded samples, standard streptavidin-biotin technique and commercially available antibodies against H3.3K27me3 and p53; fluorescence in situ hybridization (FISH) for determining *MYCN*, *EGFR*, *PDGFRA* and 1p19q status: formalin-fixed and paraffin-embedded samples and commercially available dual color FISH probes; molecular evaluations: formalin fixed and paraffin embedded samples, automated RNA extraction and PCR real-time amplification of the NTRK genes fusion panel), and gene panel (see below) was used.

The different additional analyses for each case were also chosen considering the location of tumor and the age of the patient.

### 2.3. DNA Extraction

Tumor DNAs were extracted using QIAamp Mini Kit (QIAGEN^®^, Hilden, Germany), according to the manufacturer’s instructions, and quantified using a NanoDROP 2000 Spectrophotometer (Thermo Scientific, Waltham, MA, USA).

### 2.4. Gene Panel and Bioinformatics Analysis of pHGG Panel Genes

Tumor DNA libraries were constructed using an enzymatic strategy to produce dsDNA fragments, followed by end repair, A-tailing, adapter ligation, and library amplification (Kapa Biosystems, Wilmington, MA, USA). A library pool was hybridized with SeqCap EZ Exome v3 probes (Nimblegen, Roche, Basel, Switzerland), and sequenced using NextSeq550 (Illumina Inc., San Diego, CA, USA). The reads obtained were aligned with the human reference genome hg19 using a Burrows–Wheeler Aligner (BWA, Wellcome Trust Sanger Institute, Wellcome Genome Campus, Cambridge, CB10 1SA, UK), and mapped and analyzed with IGV (Integrative Genome Viewer, IGV Broad Institute and Regents of the University of California) software.

The variant call for the identification of nucleotide variants of 34 genes (AKT1, ALK, ATRX, BRAF, CDK4, CDK6, CDKN2A, CDKN2B, EGFR, EZH2, EZHIP, H3F3A, IDH1, IDH2, KDM6A, KRAS, HRAS, NRAS, MAP2K1, MAP2K2, MLH1, MLH3, MSH2, MSH6, NF1, NF2, NTRK1, PDGFRA, PIK3CD, PIK3CA, PTEN, ROS1, TERT, TP53), was performed using automated in-house pipelines.

The paired-end reads were aligned to the human hg19 reference genome sequence using the Burrows–Wheeler Aligner v0.7.10 (BWA; http://bio-bwa.sourceforge.net, 10 November 2021, and variant calling was performed using the UnifiedGenotyper module of the Genome Analysis ToolKit v3.3-0 (GATK; https://gatk.broadinstitute.org, 9 April 2021). To discover the somatic variants present in tumor tissue, variant calling was performed using the MuTect method [39]. The final variant calling format (VCF) files were filtered and annotated using ANNOVAR software [40]. Variants that were called less than 20X, were off-target, were synonymous, or had a minor allele frequency (MAF) >5% in the Exome Aggregation Consortium (ExAC), Cambridge, MA, USA (http://exac.broadinstitute.org, 9 April 2021) were eliminated.

All variants were confirmed using Sanger sequencing. All validated variants of selected target genes were classified as pathogenic, likely pathogenic, uncertain clinical significance (VUS), uncertain clinical significance LP (VUS with minor pathogenic evidence) or benign, in agreement with the interpretation guidelines of the American College of Medical Genetics and Genomics (ACMG) [41]. Their tumor or disease association were evaluated in the somatic (Catalogue of Somatic Mutations in Cancer, COSMIC, https://cancer.sanger.ac.uk/cosmic, 14 June 2021) and constitutional (Human Gene Mutation Database, HGMD, http://www.hgmd.cf.ac.uk, 14 June 2021) databases, to clarify their hypothetical pathogenetic roles.

In silico analysis determined the pathogenicity of the variants of the splice site (BGDP https://www.fruitfly.org/seq_tools/splice.html, 14 June 2021) and the effect of the variants on the protein function (fathmm-MKL, https://fathmm.biocompute.org.uk/fathmmMKL.htm, 14 June 2021; FATHMM, http://fathmm.biocompute.org.uk/, 14 June 2021; MutationTaster, https://www.mutationtaster.org/, 14 June 2021; MutationAssessor, http://mutationassessor.org/r3/, 14 June 2021; SIFT, https://sift.bii.a-star.edu.sg/, 14 June 2021, Polyphen2_HVAR, http://genetics.bwh.harvard.edu/pph2/, 14 June 2021).

## 3. Results

Three tumors (27%) (P2, P5 and P9) are classified in the H3.3K27M mutant sub-type, two (18%) (P10 and P11) in the H3.3G34R/V mutant sub-type, one (9%) (P3) in the *IDH* mutant sub-type (Figure 1), one (9%) (P6) as a PXA-like pHGG, and one (9%) (P4) as a diffuse midline glioma *EGFR* mutant (Figure 2).

In the remaining three cases (36%) (P1, P7 and P8), the analyses carried out did not allow us to sub-classify them. (Table 1) In particular, immunohistochemistry and the successive gene panel excluded the *H3F3A* variants, the Val600Gln variant in the *BRAF* gene, and the MMR genes variants. FISH analyses executed to ascertain *MYCN*, *EGFR* and *PDGFR* amplification, and *PDGFRA* deletion, gave negative results for P1 and non-evaluable results for P7 and P8. For patient P1, considering the age and hemispheric localization of the tumor, we also ruled out the possibility of it being an infant-type pHGG through a search for fusions typical of this glioma. For patient P3, considering the presence of the *IDH2* variant and the *ATRX* wild type (Table 2), the possible presence of 1p-19q codeletion was ruled out using FISH analysis (Figure 1, Table 1).

The gene panel results are summarized in Table 2.

No variants were identified in the following target genes: *AKT1*, *ALK*, *CDK4*, *CDK6*, *CDKN2A*, *CDKN2B*, *EZH2*, *KDM6A*, *KRAS*, *NRAS*, *MAP2K1*, *MAP2K2*, *MLH1*, *MLH3*, *MSH6*, *NF2*, *NTRK1*, *PIK3CD*, *ROS1*.

All cases have at least one variant in the target genes, and most have 2–4 variants, with the exception of P3 and P11, which have nine and seven variants, respectively. Only P1 and P4 presented a single variant.

Our cases showed variants in *TP53* (six patients, 54%), *H3F3A* (six patients, 54%), *EGFR* (three patients, 27%), *PDGFRA,* and phosphatase and tensin homolog (*PTEN*) genes (two patients each, 18%). Variants in the *EZHIP*, *MSH2*, *IDH1/2*, *TERT*, *HRAS*, *NF1*, *BRAF*, *ATRX* and Phosphatidylinositol-4,5-Bisphosphate 3-Kinase Catalytic Subunit Alpha (*PIK3CA*) genes were less frequent (one patient each, 9%).

Five in six (83%) *TP53* mutated tumors present a double inactivation of the *TP53* gene (P2, P3, P8, P11 with double pathogenic variants and P10 with a pathogenic variant plus loss of heterozygosity (LOH)), while P9 shows only one hit. The *TP53* variants are five missense and four premature stop codon/frameshift (50%, respectively), localized in the core (4–8 exons) (80%) of the gene. All variants are pathogenic.

*H3F3A* variants are recurrent at the Lys27 hotspot in three in six (50%) patients (P2, P5 and P9), and in the Gly34 hotspot in two in six (33%) patients (P10 and P11). c.83A>T (p.Lys27Met) is a pathogenic variant in exon 2; instead Gly34 presents two different amino acid substitutions (c.103G>A p.Gly34Arg in P10, and c.104G>T p.Gly34Val in P11). Only one in six (16%) patients (P3) has c.246T>G (p.Asp82Glu) in exon 3 of *H3F3A* gene.

One in three (33%) *EGFR*-mutated tumors presents multiple variants (P7). None of the *EGFR* variants are benign or are distributed over the entire gene (3–20 exons).

One in two (50%) *PDGFRA*-mutated tumors showed two variants (P3).

Two in two (100%) *PTEN* mutated tumors simultaneously carried a single variant (c.302delT and c.781C>G) and the LOH of the wild-type allele of the *PTEN* gene (P10 and P11).

Unique variants are also identified in unshared genes in several patients (P1, P3, P5, P6, P9, P10 and P11).

P1 presents an uncertain significance variant (c.1328C>T p.Ser448Phe) in exon 1 of the *EZHIP* gene. P3 presents an uncertain significance variant (c.863C>T p.Ala288Val) in exon 2 of the telomerase reverse transcriptase (*TERT*) gene, a pathogenic variant in the *IDH2* gene (c.419G>A p.Arg140Gln) in exon 4, and two uncertain significance variants (c.287A>G p.Tyr96Cys and c.413G>A p.Gly138Asp) in exons 3 and 4 of the Harvey rat sarcoma viral oncogene homolog (*HRAS*). P5 shows a nonsense variant in the neurofibromatosis 1 (*NF1*) gene (c.5242C>T p.Arg1748*) in exon 37. P6 harbors a pathogenic c.1799T>A p.Val600Glu in exon 15 of the *BRAF* gene. P9 shows a benign variant (c.532G>A p.Val178Ile) in the *IDH1* gene. P10 presents an uncertain significance LP variant (c.1145G>A p.Arg382His) in exon 7 of the *MSH2* gene. P11 carries a pathogenic variant (c.2341C>T p.Arg781*) in exon 9 of the *ATRX* gene and a pathogenic variant (c.1633G>A p.Glu545Lys) in exon 10 of the *PIK3CA* gene.

## 4. Discussion

In our study, we analyzed somatic variants in eleven pHGGs in a set of target genes involved in gliomagenesis [42,43,44]. The absence of genomic DNA did not allow us to establish the exclusive somatic origin of the variants to identify a potential cancer-predisposing syndrome.

Our results confirm the literature data, indicating *TP53* and *H3F3A* as the most mutated genes in pHGGs (six patients each, 54%) [45,46] (Table 2).

The tumor suppressor and transcription factor p53 encoded by the *T53* gene plays a critical role in pHGG tumorigenesis, and its deregulation promotes cell proliferation, migration, invasion, tumor cell stemness, and the failure of apoptosis in pHGG cells. In our study, *TP53* variants were all associated with high-grade gliomas and other types of tumors (Table 2).

Variants in *H3F3A* are reported in a variety of solid tumors. We identified two recurrent pathogenic variants (Lys27Met (27%, P2, P5 and P11) and Gly34Arg (9%, P12)), a likely pathogenetic variant (Gly34Val) in one case (9%, P13), and an uncertain significance (VUS) LP variant (Asp82Glu) (9%; P3) (Table 2). c.83A>T (p.Lys27Met) is a pathogenic variant in exon 2 mainly associated with diffuse midline glioma (COSM327928). Gly34Arg and Gly34Val are pathogenic and likely-pathogenic variants associated with CNS brain tumors (COSM327929) and giant-cell tumor and astrocytoma (COSM502595), respectively. Moreover, Asp82Glu in exon 3 is reported as an uncertain significance LP, but is not present in the COSMIC database. While the role of the Lys27Met and Gly34Arg/Val variants in oncogenesis is sufficiently well-known, little is described regarding Asp82Glu. No data on frequency or specific tumors/clinical phenotype associations are reported. Interestingly Asp82 lies a region subject to post-translational modifications (two phosphorylation sites, Thr81 and Ser87, and a methylation/acetylation site, Lys80). It is not clearly known whether the substitution of aspartate with a glutamic acid, located between well-studied post-translational modification sites, could affect the final protein assembly or function. Future investigations will be indispensable in understanding the pathogenicity of Asp82Glu in the *H3F3A* gene. In two in three (67%) cases (P2 and P9) with Lys27Met we documented *TP53* variants. *TP53* variants and the consequent loss of functions are found in 70–80% of Lys27Met tumors, which contribute to radio resistance and poor prognosis [47,48].

The third most-often mutated gene in our series is *EGFR* (three cases, 27%) (Table 2). P4 shows only one pathogenic variant consisting of the insertion of nine nucleotides in exon 20 (c.2300_2301insCAGCGTGGA p.Ala767_Ser768insAlaTrpThr) in the essential C-helix domain (amino acids Ala767 to Cys775) [49]. This variant is not reported in databases but, in the same nucleotide position, different insertions are described that are associated with lung tumors [50,51]. P7 harbors VUS (c.1994G>A p.Gly665Asp and c.1973T>A p.Leu658Gln) which are exclusive to gliomas (COSM8256230 and COSM6354678, respectively). P11 harbor the VUS c.2024G>A p.Arg675Gln, which is reported in multiple tumors such as melanoma, and lung and prostate cancer, but not in brain tumors (COSM6938266). Meanwhile, it is reported in the HGMD database as a “*Disease-causing mutation?*” associated with central nervous system tumors (CM215217) [52]. The impossibility of studying the patient’s genomic DNA does not allow us to establish the germinal origin of the identified variant.

*PDGFRA* and *PTEN* were mutated in two cases (each one 18%) (Table 2).

P3 shows a likely pathogenic Val658Ala in exon 14 and a benign Gly79Asp in exon 3 of *PDGFRA*, and P6 show a VUS Gly429Arg in exon 9 of *PDGFRA*. None of these variants is reported in the HGMD. Only the benign one, Gly79Asp, has been identified in several tumors, such as prostate tumors, myelodysplastic syndrome and meningioma (COSM5019287). This variant is inserted in a complex genetic pattern of variants (9 variants) of P3, and we cannot exclude that the combinations among the variants may contribute to causing a genetic background predisposing patients to gliomagenesis.

The tumor suppressor gene *PTEN* inhibits the PI3K/AKT pathway by suppressing cellular proliferation and survival. Inactivation of the *PTEN* gene can occur in high-grade gliomas. In our study, two patients (18%) show the genetic involvement of the *PTEN* gene with complete inactivation, through the combination of a single-nucleotide variant and the LOH of the wild type allele. In particular, P10 shows a pathogenic Iso101fs variant in exon 5 not reported in COSMIC, but associated with Cowden disease (CD110165), and P11 presents an unreported VUS Gln261Glu in exon 7 (Table 2).

Variants in the *EZHIP*, *MSH2, IDH1*, *IDH2,* (Telomerase reverse transcriptase) *TERT*, *HRAS*, *NF1*, *BRAF*, *ATRX* and *PIK3CA* genes are each identified in only one patient.

P1 shows a novel VUS (c.1328C>T p.Ser443Phe) in the serine-rich region of the C-terminus of the *EZHIP* gene. *EZHIP* is a single-exon gene with an unknown function, expressed mainly in the nucleus. *EZHIP* variants occur in posterior fossa group A (PFA) ependymomas (9.4%), endometrial cancers (5%) and melanoma (2.6%) [53]. Recently, Hübner J-M et al. identified that the C-terminus of *EZHIP* is critical in the interaction with PRC2 and its inhibition [54]. Future investigations will be indispensable in understanding the role of Ser443Phe.

P10 harbors a VUS LP variant (c.1145G>A p.Arg382His) in exon 7 of *MSH2*. This variant is associated only with lung adenocarcinoma (COSM3185920) and such as *Disease-causing* mutation in colorectal cancer (CM080440) [55]. In addition, P10 exhibits biallelic inactivation of both *TP53* and *PTEN* genes, and also in this case a "sum" effect of all variants cannot be excluded.

P9 shows a benign variant (c.532G>A p.Val178Ile) in exon 6 of the *IDH1* gene, with a frequency of 5% and 194 homozygotes (ExAC_nontcga_NFE) described in association with hematopoietic neoplasms, gliomas, thyroid tumors, and meningioma (COSM97131). To strengthen the neutral effect of c.532G>A, Ward et al. reported that Val178Ile retained the wild-type ability for isocitrate-dependent NADPH production without elevating cellular 2HG levels in the cell [56]; moreover, Thirumal Kumar et al. established the neutral impact of missense Val178Ile in the IDH1 homodimer region using in silico approaches [57].

P3 presents a complex pattern of variants, and we identified a new VUS (c.863C>T p.Ala288Val) in exon 2 of the *TERT* (Telomerase reverse transcriptase) gene, as well as a pathogenic variant in exon 4 of the *IDH2* gene (c.419G>A p.Arg140Gln) associated with hematopoietic neoplasms/MDS, lung, stomach and upper aero-digestive tract tumors (COSM41590); however, these were never reported in brain tumors. Arg140 as Arg132 in the *IDH1* gene and Arg172 in *IDH2* lies in the active site of the IDH1/2 enzyme, suggesting a direct role of the variant on the catalytic niche [58]. Furthermore, P3 has two uncertain significance variants (c.287A>G p.Tyr96Cys and c.413G>A p.Gly138Asp) in exon 3 and 4 of the *HRAS* gene, not reported in COSMIC.

P5 shows a pathogenic nonsense variant, c.5242C>T p.Arg1748*, in exon 37, which is associated with NF1 (CM941094). The absence of P5 genomic DNA and specific clinical signs of NF1 does not exclude the presence of autosomal dominant NF1.

P6 has a pathogenic substitution, c.1799T>A p.Val600Glu, in exon 15 of the *BRAF* gene (COSM476). This variant has been mutated in many tumors such as melanoma, colorectal carcinoma, papillary carcinoma of thyroid and brain tumors, and Langerhans cell histiocytosis [59].

P11 has pathogenic variants in exon 9 of the *ATRX* (c.2341C>T p.Arg781*) and in exon 10 of the *PIK3CA* (c.1633G>A p.Glu545Lys) genes. c.2341C>T in *ATRX* is reported only in the COSMIC database, and is associated with melanoma and glioma (COSM1716656) instead. c.1633G>A in *PIK3CA* is reported to be associated with different tumors, including glioma (COSM763) and in megalencephaly-capillary malformation (CM126692).

For the molecular characterization of pHGGs, many integrated different diagnostic tools are required. In our study, the results derived from the gene panel were decisive for defining the molecular sub-type in two cases (P3 and P4), which were diagnosed as an uncertain sub type based on the routine diagnostic procedures (Table 1).

P3 (Figure 1) showed a pathogenic Arg140Gln variant in the *IDH2* gene which, to the best of our knowledge, has never been reported in gliomas. Raynaud et al. searched for Arg140Gln on a series of 106 gliomas, and did not find this variant in any of the gliomas studied [60]. When the *IDH2* variant occurs in gliomas, in almost all of the cases, it consists of a substitution of the arginine at codon 172, which is the exact residue analogous to Arg132 in *IDH1*. *IDH2* variants are predominantly found in the oligodendrogliomas *ATRX* and *TP53* wild type, *TERT* promoter mutated, and 1p–19q codeleted [19]. Our case is of special interest because, for the first time, Arg140Gln is documented in a glioma and occurred in a lesion with hybrid molecular features between astrocytoma and oligodendroglioma (with an absence of the *ATRX* variant, presence *TP53* variants, and no 1p–19q codeletion) (Figure 1). It remains to be ascertained whether Arg140Gln in gliomas is just an anecdotal observation, whether it is typical in children, and whether it is associated with the same histotypes and molecular features as the most common *IDH2* variants; lastly, its eventual diagnostic and prognostic significanceremains to be determined.

For patient P4 (Figure 2) the thalamic location and the age of the patient initially suggested a diagnosis of *H3F3A*-mutated gliomas. However, immunohistochemistry failed to stain for H3.3K27M-mutant protein, and successively, the gene panel analysis did not show any *H3F3A* variants. Consequently, this hypothesis was ruled out. Intriguingly, the immunostaining result for H3.3K27me3 was negative (Figure 2). FISH excluded *MYCN*, *EGFR* and *PDGFR* amplifications and *CDKN2A* deletion, while immunohistochemistry, followed by a gene panel, ruled out PXA-like pHGGs (no *BRAF* variants) (Table 2). A gene panel analysis, instead, documented an *EGFR* variant. Therefore, we diagnosed this case as a diffuse midline glioma *EGFR* mutant. This pHGG sub type is characterized by *EZHIP* epigenetic overexpression that, in turn, determines the loss of H3.3K27me3. Consequently, immunohistochemical demonstration of H3.3K27me3 loss is an effective method to indirectly ascertain its typical EZHIP overexpression [61].

Patients P1, P7 and P8 remained unclassified according to our integrated approach (Table 1).

Many causes can lead to a failure in the molecular classification of pHGGs, including the non-availability of all the necessary diagnostic tools (i.e., whole-genome methylation profiling) and the non-optimal quality of the tissues. In addition, some tumors could not fall within the currently described and classified molecular sub-types. The availability of molecular analyses reduces the number of unclassified cases.

## 5. Conclusions

*TP53* and *H3F3A* were the most mutated genes. A novel variant in *EZHIP* and an *IDH2* variant never reported in gliomas were documented. 

The gene panel was crucial in recognizing the molecular features, determining the diagnosis in two cases (an *IDH2* pathogenic variant in P3, and an *EGFR* variant in P4).

More than one-third of patients carry a variant in a gene associated with constitutional tumor-predisposing syndromes (*PTEN*, *MSH2* and *NF1*).

## Figures and Tables

**Figure 1 genes-13-00624-f001:**
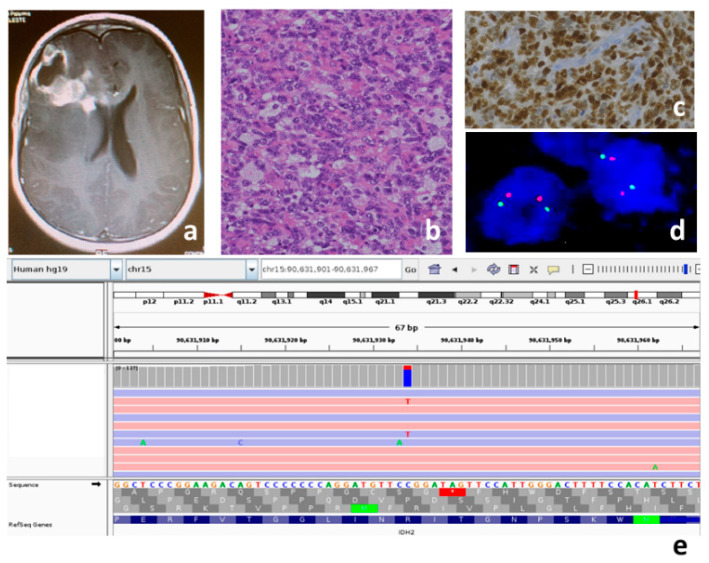
Patient P3, *IDH* mutant pHGG: (**a**) MRI T1 contrast-weighted; right frontal lesion, unevenly capturing contrast, extensive periwound edema; the tumor is in contact with the frontal horn of the right lateral ventricle; (**b**) medium-sized atypical cells and intralesional histiocytes, hematoxylin and eosin, original magnification 20×; (**c**) positive nuclear immunostaining for p53 protein as indirect expression of *TP53* mutation, original magnification 20×; (**d**) no 1p36 deletion in dual color FISH analysis (two orange and two red signals in each nucleus); (**e**) the variant c.419G>A p.Arg140Gln in exon 4 of the *IDH2* gene (NM_002168) is read in both the strands 119X (94X in G *ref* allele and 23X in T *alt* allele).

**Figure 2 genes-13-00624-f002:**
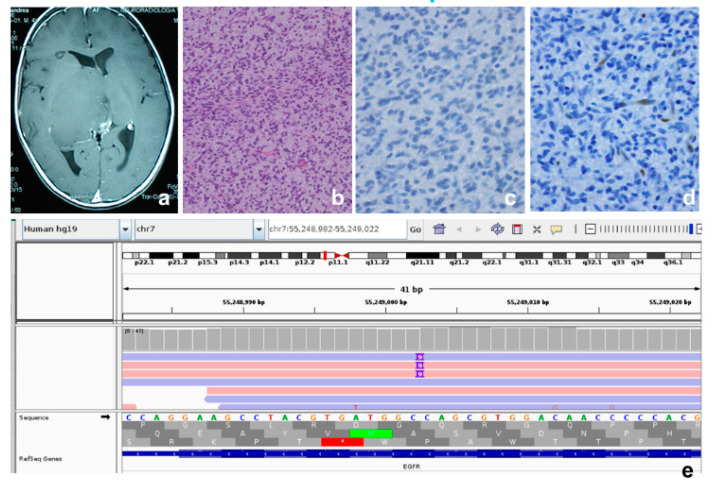
Patient P4, thalamic diffuse EGFR mutant pHGG: (**a**) T2-weighted image, presence of hyperintense mass in the right thalamic area. The tumor is in contact with the medial portion of the third ventricle and the occipital horns. No development of hydrocephalus; (**b**) spindle-shaped atypical cells, hematoxylin and eosin, original magnification 20×; (**c**) negative nuclear staining for H3.3K27M-mutant protein, original magnification 20×; (**d**) negative nuclear staining for H3.3K27me3 in tumor cells and positive control in the endothelial cells, original magnification 20×; (**e**) the variant c.2300_2301insCAGCGTGGA p.Ala767_Ser768insAlaTrpThr in exon 20 of the *EGFR* gene (NM_005228) is read in both the strands 42X.

**Table 1 genes-13-00624-t001:** Clinical data and molecular classification. F: female; M: male.

Case	Age (Years)	Sex	Localization	Molecular Classification
P1	0	F	Hemispheric	Not determined
P2	9	M	Thalamus	H3.3K27 mutant
P3	12	F	Hemispheric	*IDH* mutant
P4	4	M	Thalamus	*EGFR* mutant
P5	8	F	C3–C4	H3.3K27 mutant
P6	5	M	Hypothamalus	PXA-like
P7	10	F	Hemispheric	Not determined
P8	8	F	Cerebellum	Not determined
P9	13	M	Midbrain	H3.3K27 mutant
P10	18	M	Hemispheric	H3.3G34 mutant
P11	14	F	Hemispheric	H3.3G34 mutant

**Table 2 genes-13-00624-t002:** Variants identified by gene panel analysis. Uncertain Significance LP: VUS with minor pathogenic evidence; LOH: Loss of Heterozygosity; allele frequency from gnomAD (https://gnomad.broadinstitute.org/, 28 February 2022 NFE-population; *: HGVS (https://varnomen.hgvs.org/, 14 June 2021) stop codon.

Case	Genes	Transcript	cDNA	Protein	Exon	Allele Frequency	Variant Coverage	ACMG Classification	COSMIC
P1	*EZHIP*	NM_203407	c.1328C>T	p.Ser443Phe	1	1.09 × 10^−^^5^	119X	Uncertain Significance	-
P2	*H3F3A*	NM_002107	c.83A>T	p.Lys27Met	2	-	150X	Pathogenic	COSM327928
	*TP53*	NM_000546	c.537T>G	p.His179Gln	5	-	41X	Pathogenic	COSM11249
	*TP53*	NM_000546	c.329G>T	p.Arg110Leu	4	-	44X	Pathogenic	COSM10716
P3	*H3F3A*	NM_002107	c.246T>G	p.Asp82Glu	3	-	90X	Uncertain Significance LP	-
	*HRAS*	NM_005343	c.287A>G	p.Tyr96Cys	3	-	57X	Uncertain Significance	-
	*HRAS*	NM_005343	c.413G>A	p.Gly138Asp	4	-	83X	Uncertain Significance	-
	*IDH2*	NM_002168	c.419G>A	p.Arg140Gln	4	-	119X	Pathogenic	COSM41590
	*PDGFRA*	NM_006206	c.1973T>C	p.Val658Ala	14	-	67X	Likely Pathogenic	-
	*PDGFRA*	NM_006206	c.236G>A	p.Gly79Asp	3	8.91 × 10^−3^	145X	Benign	COSM5019287
	*TERT*	NM_198253	c.863C>T	p.Ala288Val	2	4.86 × 10^−5^	53X	Uncertain Significance	-
	*TP53*	NM_000546	c.541C>T	p.Arg181Cys	5	-	24X	Pathogenic	COSM11090
	*TP53*	NM_000546	c.916C>T	p.Arg306 *	8	-	97X	Pathogenic	COSM10663
P4	*EGFR*	NM_005228	c.2300_2301ins9	p.Ala767_Ser768ins3	20	-	42X	Pathogenic	-
P5	*H3F3A*	NM_002107	c.83A>T	p.Lys27Met	2	-	119X	Pathogenic	COSM327928
	*NF1*	NM_000267	c.5242C>T	p.Arg1748 *	37	-	83X	Pathogenic	-
P6	*BRAF*	NM_004333	c.1799T>A	p.Val600Glu	15	3.98 × 10^−6^	171X	Pathogenic	COSM476
	*PDGFRA*	NM_006206	c.1285G>A	p.Gly429Arg	9	3.43 × 10^−4^	55X	Uncertain Significance	-
P7	*EGFR*	NM_005228	c.1994G>A	p.Gly665Asp	17	-	53X	Uncertain Significance	COSM8256230
	*EGFR*	NM_005228	c.1973T>A	p.Leu658Gln	17	-	64X	Uncertain Significance LP	COSM6354678
P8	*TP53*	NM_000546	c.1047_1048ins13	p.Leu350_K351ins *	10	-	52X	Pathogenic	-
	*TP53*	NM_000546	c.451C>A	p.Pro151Thr	5	-	41X	Pathogenic	COSM43911
P9	*H3F3A*	NM_002107	c.83A>T	p.Lys27Met	2	-	155X	Pathogenic	COSM327928
	*IDH1*	NM_005896	c.532G>A	p.Val178Ile	6	4.95 × 10^−2^	111X	Benign	COSM97131
	*TP53*	NM_000546	c.817C>T	p.Arg273Cys	8	1.20 × 10^−5^	44X	Pathogenic	COSM10659
P10	*H3F3A*	NM_002107	c.103G>A	p.Gly34Arg	2	-	100X	Pathogenic	COSM327929
	*MSH2*	NM_000251	c.1145G>A	p.Arg382His	7	7.95 × 10^−6^	97X	Uncertain Significance LP	COSM3185920
	*PTEN*	NM_000314	c.302delT	p.Iso101fs (LOH)	5	-	69X	Pathogenic	-
	*TP53*	NM_000546	c.574C>T	p.Gln192 * (LOH)	6	-	79X	Pathogenic	COSM10733
P11	*ATRX*	NM_00489	c.2341C>T	p.Arg781 *	9	-	130X	Pathogenic	COSM1716656
	*EGFR*	NM_005228	c.2024G>A	p.Arg675Gln	17	2.11 × 10^−4^	540X	Uncertain Significance	COSM6938266
	*H3F3A*	NM_002107	c.104G>T	p.Gly34Val	2	-	26X	Pathogenic	COSM502595
	*PIK3CA*	NM_006218	c.1633G>A	p.Glu545Lys	10	-	81X	Pathogenic	COSM763
	*PTEN*	NM_000314	c.781C>G	p.Gln261Glu (LOH)	7	-	35X	Uncertain Significance LP	-
	*TP53*	NM_000546	c.637C>T	p.Arg213 *	6	-	152X	Pathogenic	COSM10654
	*TP53*	NM_000546	c.1024C>T	p.Arg342 *	10	-	106X	Pathogenic	COSM11073

## Data Availability

Not applicable.

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
