# Peer review of "Pediatric High Grade Glioma Classification Criteria and Molecular Features of a Case Series"

_genes, 2022, doi:10.3390/genes13040624_

Round 1
Reviewer 1 Report
This manuscript presents targeted exome sequencing results from 11 pediatric high grade gliomas. the study is observational. though the authors make mention of a classification scheme, none is proposed, nor is one justified based on these results. the methods used appear to be appropriate, but the description of methods is inadequate. the presented results should included allele frequency, but otherwise the results and discussion are interesting and pertinent. the conclusions section needs to be re-written, and the final paragraph removed. references can be improved. the appropriate literature is included, but several statements are uncited. for example, the first sentence.
Author Response
Reviewer 1:
- We mentioned the classification schemes reported in the current scientific publications;
- We improved the description of the methods of the routinely diagnostic tools used to better characterize the tumors;
- We added the allelic frequencies of the variants identified in Table 2
- We rewritten and shortened the conclusion section; the final paragraph has been removed;
- We have improved the distribution of references.
Reviewer 2 Report
The authors present case report (case series) on pediatric high grade glioma classification. The sample size is not large, it limits the conclusion. The material is novel, presenting original data. Overall, I recommend some minor revision.
English spelling check by a native speaker would be recommended.
Remarks:
Line 15: ‘mono institutional’ - could change the wording or follow standard style. I see ‘mono-institutional’ in the text.
Line 16: ‘eleven pHGGs...’ - add word ‘samples’. 11 is a small number. Add wording ‘not related’ to show that the sample collection is independent. Assume the patients were not related to each other? Here is genetical estimate, need describe sample collection.
‘routinely diagnostic tools...’ - need name these tools. ‘routine’ might differ in the clinics, need be precise in the methods description.
Line 20: ‘infrequent’ - it is only one patient, this word seems not appropriate. Write ‘relatively infrequent’ or like that.
Line 23: ‘Further studies are necessary...’ - this is too common phrase, it gives no information.
Rewrite it, or remove. Or use a phrase on the findings in the Abstract from main text Conclusion section.
Line 26: keywords - add keyword ‘case report’, change ‘Arg140Gln’ to gene ** mutation , or write more specific. The keywords should recognizable.
Line 32: ‘1.1–1.78 per 100,000’ - this sentence need a reference - where from these data?
Line 35: ‘. [1–4]’ - bulk citation. It is better to rephrase, not cite more than 1-2 reference in time. Put the dot ‘.’ after the references in parentheses [ ].
Line 52-54: ‘In the eukaryotic cell nuclei, histones...’ - it is too common phrase from textbooks about nucleosomes. May remove, rewrite, or rephrase more specific for the disease studied.
Line 75: ‘yet to be determined’ - this phrase is not complete. Maybe ‘is not known from the existing literature’. What is the hypothesis about this mutation? Rephrase, or give citation, at least.
Line 114: ‘. [20–24]’ - it is bulk citation again (5 references together - cite separately, 1-2 in time).
Line 203: ‘1.9. AIM’ - should be in lowercase ‘Aim’. Then in line 210: ‘Our goal’ - use word ‘aim’. Goal is to study genetics mechanisms of glioma, find novel gene markers.
Please describe the sample here - did all the patients not related genetically?
Line 225: ‘-FISH- ’ - use parentheses instead of hyphen line. May not show abbreviation FISH, it is quite standard.
Line 227: ‘analyses executed’ - rewrite (English grammar is not proper).
Line 242 and below: ‘variants of AKT1,...’ - cunt number of genes, describe the pipeline in short.
Line 252, 253 - ‘COSMIC’, ‘HGMD’ - add references for these databases (web-links)
Line 255: ‘splice site (BGDP http://www.fruitfly.org/)’ - this is not correct link, on the Drosophila database. Add correct link to the splice sites analysis.
Line 256: ‘fathmm -MKL, FATHMM, MutationTaster, MutationAssessor, SIFT, Polyphen2_HVAR’ - need cite these tools, by references or web-links.
The bioinformatics part of the manuscript could be written in more details.
May use references on glioma bioinformatics from these recent papers:
https://pubmed.ncbi.nlm.nih.gov/33635875/
Line 263: ‘. (Table 1)’ - typo.
Figure 1 and 2 - the panel letters (a), (b) ... - could be before the sentences, not after. It is hard to read. Need some comments in the text, what one can see in the figure panels. Figure 1 and 2 should be separated by some text between them. Put it on different pages
Avoid abbreviations like P3 in the figure legend - write in full Patient P3.
Try format Table 2 to fit one page (change font, spaces between lines, columns)
Line 335: ‘[41–43]’ - bulk citation again.
Description of all the gene variants in the Discussion section looks repeated. May add a figure, or scheme instead of each variant description, to make it visually presented. This is on the authors’ discretion.
Overall technical remarks - the manuscript is not formatted properly
Some standard sections are not filled
‘Citation: Lastname, F.’
At the end of the text (lines 484 and below) - fill sections
‘Author Contributions:’ etc.
Author Response
Reviewer 2:
- “Line 15”: we changed the word according to the suggestion;
- “Line 16”: we added the word “samples” and specified “not related”;
- We improved the description of the methods of the routinely diagnostic tools used to better characterize the tumors;
- “Line 20”: we wrote “relatively infrequent”;
- “Line 23”: we removed the sentence;
- “Line 26”: we added “case report” and changed “Arg140Gln” into “mutation”;
- “Line 32”: we added a reference;
- “Line 35”, “Line 114” and “Line 335”: When possible we reduced the number of the references and put a dot after the references in parentheses;
- “Lines 52-54”: we rewrote the phrase;
- “Line 75 (76)”: we removed the phrase “yet to be determined”;
- “Line 203”: we changed “AIM” in “Aim” and “Our goal” in “Our aim”;
- We specified that the patients were not related genetically;
- “Line 225”: we used parentheses;
- “Line 227” (228): we removed the word “executed”;
- “Line 242 and below”: we counted the number of gene and we wrote “The variant call for identification of nucleotide variants of 34 genes (AKT1, ALK, ATRX, BRAF, CDK4, CDK6, CDKN2A, CDKN2B, EGFR, EZH2, EZHIP, H3F3A, IDH1, IDH2, KDM6A, KRAS, HRAS, NRAS, MAP2K1, MAP2K2, MLH1, MLH3, MSH2, MSH6, NF1, NF2, NTRK1, PDGFRA, PIK3CD, PIK3CA, PTEN, ROS1, TERT, TP53)” and we detailed our pipeline;
- “Lines 252, 253”: we added web links to the COSMIC and HGMD databases;
- “Line 255”: we added a correct web links of BGDP;
- “Line 256”: we added web-links to the bioinformatic tools;
- We detailed our bioinformatics pipeline without using the reference document (pubmed 33635875) because it referred to an expression study and not a gene panel study to identify genomic variants;
- “Line 263” (Table): we corrected the typo;
- Figure 1 and 2 legends: we moved the panel letters before the sentences; in the Discussion section we have added a reference to figure 1 (the one for figure 2 was already there) where we describe the tumors of figures 1 and 2; we separated the figures by some text between them; we put the two figures on different pages;
- We added “Patient” before P3 and P4 in the figure legends;
- We tried to format Table 2 to fit one page but it is a bit difficult. We asked to the Editor to help us to fit the Table 2 in a one page changing font and spaces between lines and columns;
- We described the genetic variants in discussion without a table or schema because we added a specific comment to each variant;
- We checked the formatting particularly in the References section;
- “Line 484 and below”: we filled sections;
- We completed the author contributions etc.
Reviewer 3 Report
Pediatric high grade gliomas classification criteria and molecular features of a case series
To study the molecular subtypes of pediatric high-grade gliomas, Buccoliero et al. studied 11 patient samples and found novel somatic mutations in a targeted gene panel.
Major comments:
- How these subtypes correlate with the prognosis of the patients?
- The sample size is small, making the interpretations less conclusive. How is the finding contributing to the diagnosis of the patients?
Author Response
Reviewer 3:
- We have not investigated the possible correlations between molecular characteristics and prognosis. In fact, although it would be interesting information, we believe that due to the small size of our series, any results obtained would have been questionable.